# Peer review of "Older People’s Help-Seeking Behaviors in Rural Contexts: A Systematic Review"

_ijerph, 2022, doi:10.3390/ijerph19063233_

Round 1
Reviewer 1 Report
Brief summary
This is an interesting study featuring a review which is referred to the conditions of older people's HSBs who are living in rural and remote areas. An important clarification of evidence about HSBs took part in relation to other examined factors as far as the rural health field.
Broad comments
Article ananyses successfully the obstacles that elderly populations face regarding HSB and examines the design issues for interventions to improve citizens' health and quality of life in rural communities. This systematic review has a mercurial type, maybe the results section should be more matched in different categorizes or with other structure.This study contributes positively as a quide for local authorities as far as the rural planning based on various future regulations or rural development plans. Process also minds specific sociοeconomic characteristics and the existing patients' inequalities of rural context.
Author Response
Responses to the reviewers’ comments
Thank you very much for reviewing our manuscript and providing suggestions for its improvement. We have provided point-by-point responses to the reviewers’ comments; our revisions are indicated in red font in the manuscript. We hope that the revised manuscript meets the journal’s requirements and can now be considered for publication.
Brief summary
This is an interesting study featuring a review which is referred to the conditions of older people's HSBs who are living in rural and remote areas. An important clarification of evidence about HSBs took part in relation to other examined factors as far as the rural health field.
Broad comments
Article analyses successfully the obstacles that elderly populations face regarding HSB and examines the design issues for interventions to improve citizens' health and quality of life in rural communities. This systematic review has a mercurial type, maybe the results section should be more matched in different categorizes or with other structure. This study contributes positively as a guide for local authorities as far as the rural planning based on various future regulations or rural development plans. Process also minds specific socioeconomic characteristics and the existing patients' inequalities of rural context.
Response:
Thank you for your valuable feedback. We have revised the result sections comprehensively based on the feedback from you and the other reviewers, for consistency.
Reviewer 2 Report
The paper properly analyzes important issues for present conditions of older people’s HSBs in rural contexts.
The article is well structured. However, It needs to provide the theoretical background of the study model.
The study should provide more results of older people’s HSBs in rural area based on the previous study.
In Introduction: It needs to address the research trend in the field referring previous studies.
In the Literature Review: There is currently no literature review. It needs to provide theoretical background of older people’s HSBs in rural contexts..
The methods are adequately described.
Author Response
Responses to the reviewers’ comments
Thank you very much for reviewing our manuscript and providing suggestions for its improvement. We have provided point-by-point responses to the reviewers’ comments; our revisions are indicated in red font in the manuscript. We hope that the revised manuscript meets the journal’s requirements and can now be considered for publication.
The paper properly analyzes important issues for present conditions of older people’s HSBs in rural contexts. The article is well structured. However, It needs to provide the theoretical background of the study model. The study should provide more results of older people’s HSBs in rural area based on the previous study. In Introduction: It needs to address the research trend in the field referring previous studies.
In the Literature Review: There is currently no literature review. It needs to provide theoretical background of older people’s HSBs in rural contexts.
Response:
Thank you for your valuable feedback. We have revised the introduction by adding a theoretical framework comparing between urban and rural settings, including HSBs.
The methods are adequately described.
Reviewer 3 Report
Dear authors,
I read the manuscript with great interest.
As many countries of the world are aging, the content of this review will be useful in many parts of the world.
[Introduction]
>Although the Internet and social media have made knowledge gathering easier, they are primarily used by young and middle-aged individuals
 This explanes that young and middle-aged people use the Internet. However, I think it is necessary to show what percentage of the Internet usage rate of elderly.
>Older people tend to obtain information from television and newspapers [14,15].
 I think that the content of "Older people tend to obtain information from television and newspapers" is not written in either the 14th or 15th papers. The paper to be cited might be incorrect.
I think it is better to check the other citations once again for any mistakes.
If possible you should better write a specific purpos. Readers may expect more specific purposes. Because the purpose is abstract, I couldn't grasp the big picture of this review until I read the discussion.
However, this view is an exploratory summary of the dissertation content, so it may be difficult to write a specific purpose.
[Method]
What keywords did you use for your search? I think it's a good idea to include keywords to help readers understand this review.
[Result]
Table 3
Why did you tabulate the country breakdown? I think that what is written in the results should be mentioned in the discussion.
[Conclusion]
I think the conclusion should contain the claims of this department.
> This systematic review clarified the current evidence regarding HSBs among older people living in rural areas.
What is "the current evidence regarding HSBs among older people living in rural areas."?
[Citation]
The citation numbers are not in order.
Author Response
Responses to the reviewers’ comments
Thank you very much for reviewing our manuscript and providing suggestions for its improvement. We have provided point-by-point responses to the reviewers’ comments; our revisions are indicated in red font in the manuscript. We hope that the revised manuscript meets the journal’s requirements and can now be considered for publication.
Dear authors,
I read the manuscript with great interest.
As many countries of the world are aging, the content of this review will be useful in many parts of the world.
[Introduction]
>Although the Internet and social media have made knowledge gathering easier, they are primarily used by young and middle-aged individuals
 This explanes that young and middle-aged people use the Internet. However, I think it is necessary to show what percentage of the Internet usage rate of elderly.
Response:
Thank you for your valuable feedback. We have revised the introduction by adding the rates of internet usage.
>Older people tend to obtain information from television and newspapers [14,15].
 I think that the content of "Older people tend to obtain information from television and newspapers" is not written in either the 14th or 15th papers. The paper to be cited might be incorrect.
I think it is better to check the other citations once again for any mistakes.
Response:
Thank you for your valuable feedback. We have revised all of the references based on the suggestions.
If possible you should better write a specific purpos. Readers may expect more specific purposes. Because the purpose is abstract, I couldn't grasp the big picture of this review until I read the discussion. However, this view is an exploratory summary of the dissertation content, so it may be difficult to write a specific purpose.
Response:
Thank you for your valuable feedback. We have revised the background by adding the purpose of this research.
[Method]
What keywords did you use for your search? I think it's a good idea to include keywords to help readers understand this review.
Response:
Thank you for your valuable feedback. We have revised the method section by adding the search strategy with the search keywords.
[Result]
Table 3
Why did you tabulate the country breakdown? I think that what is written in the results should be mentioned in the discussion.
Response:
Thank you for your valuable feedback. We have revised the discussion part by adding description regarding the lack of evidence of HSBs, referring to a few studies from several countries.
[Conclusion]
I think the conclusion should contain the claims of this department.
> This systematic review clarified the current evidence regarding HSBs among older people living in rural areas.
What is "the current evidence regarding HSBs among older people living in rural areas."?
Response:
Thank you for your valuable feedback. We have revised the conclusion by adding concrete description of the results.
[Citation]
The citation numbers are not in order.
Response:
Thank you for your valuable feedback. We have revised all of the references based on the suggestions.
Reviewer 4 Report
Older people’s help-seeking behaviors in rural contexts: A systematic review (ijerph-1608820): Reviewer Comments
Thank you for the opportunity to review this paper, which reviews existing studies on help-seeking behaviours of older people in rural and regional areas in order to synthesise the available evidence and recommend further research approaches. This is an area of significant international interest as aging populations and their healthcare support are a timely concern in many nations globally, and is particularly problematic because of the strain COVID-19 puts on the existing healthcare systems. Thus the addition of a study bringing together what is known and what needs to be found out has real potential merit. However, I recommend the present paper be revised in order to more strongly frame what it proposes to look at and contribute in order to meet this potential. Specifically:
Introduction – “Help Seeking Behaviour” here lacks a really strong definition; this could be drawn from elsewhere, and quoted directly, but really needs something clear and defined so as to launch the paper with a really clear frame. A good definition, for example, might be drawn from Rickwood & Thomas (2012): “any action of energetically seeking help from the health care services or from trusted people in the community and includes understanding, guidance, treatment and general support”. It is also useful at this point to set out that, while HSB has traditionally been used most strongly in mental health research/policy/practice (as the paper’s own core references show) it has wider applicability to holistic understandings of health and wellbeing. Even though in Limitations it’s set out that HSB means different things to different people, and that the scope is wide, the paper would still start stronger with a good definition, even if only to then expand further on later.
Further to this, the list of what it might include in the existing definition is quite broad, and doesn’t really match up with what seems to be commonly understood as HSB in most existing literature; some of them (consulting, utilizing professional care) align with HSB while others (modifying eating habits, performing regular exercise, screening) would seem more to be outcomes of HSB than really HSB itself. With a strong definition in place, I recommend reframing what are going to be considered as HSBs in this study as it will justify the scope and the inclusion of particular areas of focus in the systematic review.
“Excessive use of medical care” – this needs further definition. Does this mean overtreatment/overprescribing on the part of healthcare practitioners working with the elderly in these areas? Or does it mean high levels, often unnecessarily, of medical treatment sought by the elderly in these areas, to the detriment of their ability to self-manage their own conditions? (see Ouchida & Lachs 2015). The information about media/internet and HSBs is relevant to both urban and rural older people; it would make more sense to be positioned before discussing the differences in rural people’s HSB. One thing that is also not mentioned here, alongside the scarcity of resources, is the other barriers that prevent rural elderly from accessing healthcare; although an older paper now, Goins et al 2005 synthesise these barriers well, including transport, isolation, financials; another barrier not mentioned is also maybe a cultural barrier to HSBs among rural adults depending on the country in question.
Methods – the description and conduct of the SR itself seems well done. Regarding the use of specific measures of Quality of Life (QOL) and Self-Rated Health (SRH) it would be good to split the mention of these two categories of the analysis into separate sentences at the end of this section. (e.g. “We also examined two further categories: Quality of Life (QOL) which is defined as…. , and Self-Rated Health (SRH) which is defined as… .) Then you could specifically define what you (and your included papers) are considering as QOL (whether qualitative or quantitative) and similarly with SRH, whether this is the standard questionnaire-style World Health Organization understanding of it.
Findings – The categorical analysis of the reviewed papers is well done. The application of sociological lenses (gender, education, SES) is good, but perhaps an added sub-section of 3.4, after “associations with backgrounds”, might be “barriers”; given that a lot of 3.4.1 covers barriers anyway it might help split/define the content of this section more clearly.
Discussion – mostly good, as good policy and practice implications and recommendations are made; however there should be more made of the fact that these things are situational and not globally applicable (for example, “rural” in the context of a huge country like Australia or the United States involves different problems to “rural” in the context of smaller countries like Ireland, Malaysia or Norway; likewise socio-economic and educational disparities have a different contextual foundation in wealthier countries than low-income countries). It would also be good to see the implications and recommendations for additional scholarly or research attention alongside the policy and practice recommendations.
Good point in Limitations that meta-analysis difficult across varied types of papers; one future spin-off alongside this traditional systematic review might be to undertake a meta-ethnography of existing qualitative studies of older rural people and health behaviours/experiences/attitudes looking for HSB themes.
References:
Goins, R. Turner, et al. "Perceived barriers to health care access among rural older adults: a qualitative study." The Journal of Rural Health 21.3 (2005): 206-213.
Ouchida, Karin M., and Mark S. Lachs. “Not for Doctors Only: Ageism in Healthcare.” Generations: Journal of the American Society on Aging 39, no. 3 (2015): 46–57. https://www.jstor.org/stable/26556135.
Rickwood D, Thomas K. Conceptual measurement framework for help-seeking for mental health problems. Psychol Res Behav Manag. 2012;5:173-83. doi: 10.2147/PRBM.S38707.
Author Response
Responses to the reviewers’ comments
Thank you very much for reviewing our manuscript and providing suggestions for its improvement. We have provided point-by-point responses to the reviewers’ comments; our revisions are indicated in red font in the manuscript. We hope that the revised manuscript meets the journal’s requirements and can now be considered for publication.
Thank you for the opportunity to review this paper, which reviews existing studies on help-seeking behaviours of older people in rural and regional areas in order to synthesise the available evidence and recommend further research approaches. This is an area of significant international interest as aging populations and their healthcare support are a timely concern in many nations globally, and is particularly problematic because of the strain COVID-19 puts on the existing healthcare systems. Thus the addition of a study bringing together what is known and what needs to be found out has real potential merit. However, I recommend the present paper be revised in order to more strongly frame what it proposes to look at and contribute in order to meet this potential. Specifically:
Introduction – “Help Seeking Behaviour” here lacks a really strong definition; this could be drawn from elsewhere, and quoted directly, but really needs something clear and defined so as to launch the paper with a really clear frame. A good definition, for example, might be drawn from Rickwood & Thomas (2012): “any action of energetically seeking help from the health care services or from trusted people in the community and includes understanding, guidance, treatment and general support”. It is also useful at this point to set out that, while HSB has traditionally been used most strongly in mental health research/policy/practice (as the paper’s own core references show) it has wider applicability to holistic understandings of health and wellbeing. Even though in Limitations it’s set out that HSB means different things to different people, and that the scope is wide, the paper would still start stronger with a good definition, even if only to then expand further on later.
Response:
Thank you for your valuable feedback. We have revised the introduction by including the suggested definition of HSB, with the references.
Further to this, the list of what it might include in the existing definition is quite broad, and doesn’t really match up with what seems to be commonly understood as HSB in most existing literature; some of them (consulting, utilizing professional care) align with HSB while others (modifying eating habits, performing regular exercise, screening) would seem more to be outcomes of HSB than really HSB itself. With a strong definition in place, I recommend reframing what are going to be considered as HSBs in this study as it will justify the scope and the inclusion of particular areas of focus in the systematic review.
Response:
Thank you for your valuable feedback. We have revised the third paragraph of the background by adding this study’s focus on HSB.
“Excessive use of medical care” – this needs further definition. Does this mean overtreatment/overprescribing on the part of healthcare practitioners working with the elderly in these areas? Or does it mean high levels, often unnecessarily, of medical treatment sought by the elderly in these areas, to the detriment of their ability to self-manage their own conditions? (see Ouchida & Lachs 2015). The information about media/internet and HSBs is relevant to both urban and rural older people; it would make more sense to be positioned before discussing the differences in rural people’s HSB. One thing that is also not mentioned here, alongside the scarcity of resources, is the other barriers that prevent rural elderly from accessing healthcare; although an older paper now, Goins et al 2005 synthesise these barriers well, including transport, isolation, financials; another barrier not mentioned is also maybe a cultural barrier to HSBs among rural adults depending on the country in question.
Response:
Thank you for your valuable feedback. We have revised the third paragraph by adding the definition of excessive usage of health care and description regarding accessibility and availability of medical care as barriers, referring to the suggested article.
Methods – the description and conduct of the SR itself seems well done. Regarding the use of specific measures of Quality of Life (QOL) and Self-Rated Health (SRH) it would be good to split the mention of these two categories of the analysis into separate sentences at the end of this section. (e.g. “We also examined two further categories: Quality of Life (QOL) which is defined as…. , and Self-Rated Health (SRH) which is defined as… .) Then you could specifically define what you (and your included papers) are considering as QOL (whether qualitative or quantitative) and similarly with SRH, whether this is the standard questionnaire-style World Health Organization understanding of it.
Response:
Thank you for your valuable feedback. We have added the definitions of QOL and SRH based on the references.
Findings – The categorical analysis of the reviewed papers is well done. The application of sociological lenses (gender, education, SES) is good, but perhaps an added sub-section of 3.4, after “associations with backgrounds”, might be “barriers”; given that a lot of 3.4.1 covers barriers anyway it might help split/define the content of this section more clearly.
Response:
Thank you for your valuable feedback. We have added the subsection 3.4.2 “Barriers,” consisting of the second paragraph to the original subsection 3.4.1.
Discussion – mostly good, as good policy and practice implications and recommendations are made; however there should be more made of the fact that these things are situational and not globally applicable (for example, “rural” in the context of a huge country like Australia or the United States involves different problems to “rural” in the context of smaller countries like Ireland, Malaysia or Norway; likewise socio-economic and educational disparities have a different contextual foundation in wealthier countries than low-income countries). It would also be good to see the implications and recommendations for additional scholarly or research attention alongside the policy and practice recommendations.
Response:
Thank you for your valuable feedback. We have added the implications of this research to the future studies in the discussion section.
Good point in Limitations that meta-analysis difficult across varied types of papers; one future spin-off alongside this traditional systematic review might be to undertake a meta-ethnography of existing qualitative studies of older rural people and health behaviours/experiences/attitudes looking for HSB themes.
Response:
Thank you for your valuable feedback. We have added the reviewer’s suggestions to the limitations part in the discussion.
References:
Goins, R. Turner, et al. "Perceived barriers to health care access among rural older adults: a qualitative study." The Journal of Rural Health 21.3 (2005): 206-213.
Ouchida, Karin M., and Mark S. Lachs. “Not for Doctors Only: Ageism in Healthcare.” Generations: Journal of the American Society on Aging 39, no. 3 (2015): 46–57. https://www.jstor.org/stable/26556135.
Rickwood D, Thomas K. Conceptual measurement framework for help-seeking for mental health problems. Psychol Res Behav Manag. 2012;5:173-83. doi: 10.2147/PRBM.S38707.
Reviewer 5 Report
This manuscript is an interesting contribution that helps to clarify a still hazy and poorly explored area of community care; I really appreciate that the authors have given relevance to the care of the elderly population in the rural context.
The article is well structured, the information contained is complete and accurate, and the hypothesis is verifiable; the revision complies with the best international standards.
There are some inaccuracies within the text and in the tables listed below:
- Table 2: check that the words in the header are not broken (e.g. country).
- Table 2: verify the results of the "Xu [2020] [33]" study; the table states "The factors associated with self-treatment were health status, recent alcohol consumption, and the utilization of family practice services", but this sentence is unclear. In the original article, Xu et al. (2020) state "Better health status, no recent alcohol consumption and no utilization of family practice are associated with self-treatment among rural elders." I invite the authors to rephrase the sentence, for example by stating "The factors associated with self-treatment were better health status, no recent alcohol consumption and no utilization of family practice".
- Table 3: missing "Number of studies" in the header.
- Table 4: the number of columns must be consistent with table 3; both tables must either have 3 columns (Variable, Number of studies and %) or both 2 columns (Variable and Number of studies (%)).
- Table 4: the percentage value (75%) is missing in the "Content of HSBs - Professional care" line.
- Page 10: the phrase "Health conditions also reduced the use of self-care. Good health conditions and alcohol consumption were both related to self-treatment for symptoms [33]" is unclear (see annotation regarding Xu et al., 2020); I recommend rewriting the sentence to avoid misunderstandings about alcohol consumption.
- Title 3.4.4.: write "self-rated health" in full before "SRH" (e.g. self-rated health (SRH)).
- Conclusions: "[...] there is a lack of evidence regarding the relationship between HSBs and QOL in rural contexts over long periods of time, measured through longitudinal studies." Evidence is also lacking for Diagnosis, Trend for professional care, Self-rated health and Mortality, as reported in table 4; I invite the authors to integrate these considerations into the conclusions.
I remain at your disposal for further clarifications. I would like to congratulate the authors on the rigor with which they drafted this systematic review.
Best regards.
Author Response
Responses to the reviewers’ comments
Thank you very much for reviewing our manuscript and providing suggestions for its improvement. We have provided point-by-point responses to the reviewers’ comments; our revisions are indicated in red font in the manuscript. We hope that the revised manuscript meets the journal’s requirements and can now be considered for publication.
This manuscript is an interesting contribution that helps to clarify a still hazy and poorly explored area of community care; I really appreciate that the authors have given relevance to the care of the elderly population in the rural context.
The article is well structured, the information contained is complete and accurate, and the hypothesis is verifiable; the revision complies with the best international standards.
There are some inaccuracies within the text and in the tables listed below:
- Table 2: check that the words in the header are not broken (e.g. country).
Response:
Thank you for your valuable feedback. We have revised Table 2 comprehensively based on the feedback.
- Table 2: verify the results of the "Xu [2020] [33]" study; the table states "The factors associated with self-treatment were health status, recent alcohol consumption, and the utilization of family practice services", but this sentence is unclear. In the original article, Xu et al. (2020) state "Better health status, no recent alcohol consumption and no utilization of family practice are associated with self-treatment among rural elders." I invite the authors to rephrase the sentence, for example by stating "The factors associated with self-treatment were better health status, no recent alcohol consumption and no utilization of family practice".
Response:
Thank you for your valuable feedback. We have revised the explanation based on the feedback and review of the research.
- Table 3: missing "Number of studies" in the header.
Response:
Thank you for your valuable feedback. We have added the suggested phrase to Table 3.
- Table 4: the number of columns must be consistent with table 3; both tables must either have 3 columns (Variable, Number of studies and %) or both 2 columns (Variable and Number of studies (%)).
Response:
Thank you for your valuable feedback. We have added the suggested phrase to Table 4.
- Table 4: the percentage value (75%) is missing in the "Content of HSBs - Professional care" line.
Response:
Thank you for your valuable feedback. We have added the suggested phrase to Table 4.
- Page 10: the phrase "Health conditions also reduced the use of self-care. Good health conditions and alcohol consumption were both related to self-treatment for symptoms [33]" is unclear (see annotation regarding Xu et al., 2020); I recommend rewriting the sentence to avoid misunderstandings about alcohol consumption.
Response:
Thank you for your valuable feedback. We have revised the sentence based on the suggestion and the reference.
- Title 3.4.4.: write "self-rated health" in full before "SRH" (e.g. self-rated health (SRH)).
Response:
Thank you for your valuable feedback. We have revised all of the abbreviations.
- Conclusions: "[...] there is a lack of evidence regarding the relationship between HSBs and QOL in rural contexts over long periods of time, measured through longitudinal studies." Evidence is also lacking for Diagnosis, Trend for professional care, Self-rated health and Mortality, as reported in table 4; I invite the authors to integrate these considerations into the conclusions.
Response:
Thank you for your valuable feedback. We have revised the conclusion by adding concrete description of the results.
I remain at your disposal for further clarifications. I would like to congratulate the authors on the rigor with which they drafted this systematic review.
Best regards.
Round 2
Reviewer 4 Report
Thank you to the authors for engaging so genuinely and diligently with review feedback. All the points of revision I previously identified have been adequately responded to, and I am happy to recommend this paper proceed to publication.